# The association between low-level viraemia and subsequent viral non-suppression among people living with HIV/AIDS on antiretroviral therapy in Uganda

**Nicholus Nanyeenya**[1,2]*, **Larry William Chang**[3], **Noah Kiwanuka**[1], **Esther Nasuuna**[4], **Damalie Nakanjako**[5], **Gertrude Nakigozi**[6], **Simon P. S. Kibira**[7], **Susan Nabadda**[2], **Charles Kiyaga**[2], **Fredrick Makumbi**[1]

1 Department of Epidemiology and Biostatistics, School of Public Health, Makerere University College of Health Sciences, Kampala, Uganda, 2 Department of National Health Laboratory Services, Ministry of Health, Kampala Uganda, 3 Department of Epidemiology, School of Medicine, Johns Hopkins University, Baltimore, Maryland, United States of America, 4 Infectious Diseases Institute, Makerere University College of Health Sciences, Kampala, Uganda, 5 Department of Medicine, School of Medicine Makerere University College of Health Sciences, Kampala, Uganda, 6 Rakai Health Sciences Project, Rakai, Uganda, 7 Department of Community Health and Behavioral Sciences, School of Public Health, Makerere University College of Health Sciences, Kampala, Uganda

* nanyeenya@gmail.com

**Data Availability Statement:** All relevant data are within the paper and its Supporting Information files.

## Abstract

### Background

Uganda's efforts to end the HIV epidemic by 2030 are threatened by the increasing number of PLHIV with low-level viraemia (LLV). We conducted a study to determine the prevalence of LLV and the association between LLV and subsequent viral non-suppression from 2016 to 2020 among PLHIV on ART in Uganda.

### Method

This was a retrospective cohort study, using the national viral load (VL) program data from 2016 to 2020. LLV was defined as a VL result of at least 50 copies/ml, but less than 1,000 copies/ml. Multivariable logistic regression was used to determine the factors associated with LLV, and cox proportional hazards regression model was used to determine the association between LLV and viral non-suppression.

### Results

A cohort of 17,783 PLHIV, of which 1,466 PLHIV (8.2%) had LLV and 16,317 (91.8%) had a non-detectable VL was retrospectively followed from 2016 to 2020. There were increasing numbers of PLHIV with LLV from 2.0% in 2016 to 8.6% in 2020; and LLV was associated with male sex, second line ART regimen and being of lower age. 32.5% of the PLHIV with LLV (476 out of 1,466 PLHIV) became non-suppressed, as compared to 7.7% of the PLHIV (1,254 out of 16,317 PLHIV) with a non-detectable viral load who became non-suppressed during the follow-up period. PLHIV with LLV had 4.1 times the hazard rate of developing

**Funding:** This project was supported by NIH Research Training Grant # D43TW009340 funded by the NIH Fogarty International Center, NINDS and NIMH. The content is solely the responsibility of the authors and does not necessarily represent the official views of the National Institutes of Health.

**Competing interests:** The authors have declared that no competing interests exist.

**Abbreviations:** AIDS, Acquired Immune Deficiency Syndrome; ART, Antiretroviral Therapy; CDC, Centres for Disease Control and Prevention; CPHL, Central Public Health Laboratories; DBS, Dried Blood Spot; HIV, Human Immunodeficiency Virus; IAC, Intensive Adherence Counselling; IAPAC, International Association of Providers of AIDS Care; LLV, Low-level viraemia; PLHIV, People Living with HIV/AIDS; VL, Viral Load; WHO, World Health Organization.

viral non-suppression, as compared to PLHIV with a non-detectable VL (adjusted hazard ratio was 4.1, 95% CI: 3.7 to 4.7, p < 0.001).

## Conclusion

Our study indicated that PLHIV with LLV increased from 2.0% in 2016 to 8.6% in 2020, and PLHIV with LLV had 4.1 times the hazard rate of developing viral non-suppression, as compared to PLHIV with a non-detectable VL. Hence the need to review the VL testing algorithm and also manage LLV in Uganda.

## Introduction

The introduction of antiretroviral therapy (ART) in the late 1980's and 1990's led to a marked improvement in HIV/AIDS care across the world, transforming the originally fatal and AIDS defining HIV epidemic into a manageable chronic condition with improved quality of life [1, 2]. Numerous efforts have been devoted to scale up access to ART, and an estimated 27.5 million (out of 37.7 million) PLHIV worldwide were accessing it by the end of 2020 [3]. In Uganda, about 1.2 million (out of 1.4 million) PLHIV were accessing ART by December 2019 [4]. The increase in access to ART has simultaneously led to the scale up of HIV viral load (VL) coverage in Uganda from about 2% of PLHIV on ART in 2014 to about 95% of PLHIV on ART 2020 [5]. Uganda is currently devoting enormous efforts to achieve the global targets of ending the HIV/ AIDS epidemic by 2030 [6, 7], though the increasing concern of PLHIV with low-level viraemia (≥50 to <1,000 copies/ml) is posing a risk to this progress. This is because low-level viraemia (LLV) has been associated with viral non-suppression and virologic failure by the different studies elsewhere [8–10]. To our knowledge however, no specific study has been done to determine the association between LLV and viral non-suppression in Uganda. Similarly, we are not aware about the availability of any intervention(s) that target PLHIV with LLV in Uganda.

The goal of ART is to lead to HIV viral suppression, with an increase in the body immunity function [11]. In 2013, the World Health Orgnisation (WHO) recommended VL monitoring as the preferred way to monitor PLHIV on ART [12]. HIV viral load is the number of HIV viral RNA copies per milliliter of blood, and a threshold of 1,000 copies/ml is used to determine viral non-suppression. Hence a VL of 1,000 copies/ml or higher is crucial in early detection of either poor drug adherence or virologic treatment failure [13]. Recent studies have associated VL non-suppression with several risk factors like comorbidities, poor adherence to ART, sociodemographic and psychological factors, poor absorption of antiretroviral drugs, lack of knowledge or awareness of the benefits of viral suppression, and drug toxicity among others [14–16]. Viral non-suppression has also been associated with an increased risk of fastened progression to AIDS and poor clinical outcomes [17, 18]. Despite the WHO recommendation of using a threshold of 1,000 copies/ml [17], the Centers for Disease Control and Prevention (CDC) and the International Association of Providers of AIDS Care (IAPAC) recommend use of 200 copies/ml as a threshold for determining VL non-suppression, and this has been adopted in various developed countries [19, 20].

Uganda adopted the WHO 2013 recommendation and initiated the scale up of VL testing in 2014 by establishing a national VL testing reference laboratory at Ministry of Health Central Public Health Laboratories (CPHL) [21]. Since then, VL coverage has increased from 2% of PLHIV on ART in 2014 to 95% in 2020 [5]. Like most of other Sub-Saharan African (SSA) countries, Uganda also uses a threshold of 1,000 copies to determine VL non-suppression [22],

yet there have been concerns with this threshold that it could lead to accumulation of PLHIV with LLV, since it is considered as being very high [23].

Different studies have shown that LLV ($\geq$50 to <1,000 copies/ml) can be associated with HIV virological failure and drug resistance, which may lead to accelerated disease progression [24–27]. In a study to evaluate virologic failure and its predictors in four African countries including Uganda, LLV and persistent LLV were observed in 19.3% and 7.8% of PLHIV on ART respectively [9]. In this study, 57.5% of participants with persistent LLV (plasma HIV RNA > 50 copies/mL at two consecutive visits) later had confirmed virologic failure. In South African study, LLV was associated with increased hazards of virological failure and the subsequent switch to second-line ART, as compared with a non-dectectable VL of less than 50 copies/ml; and the risk of virological failure was increased more with increased ranges of LLV [10]. However there is limited data about the association of LLV and viral non-suppression in Uganda, yet this is very important to guide key policy decisions in the country.

In this study, we aimed to determine the association between low-level viraemia and subsequent viral non-suppression among PLHIV on ART in Uganda.

## Methods

### Study design and population

This was a retrospective cohort study involving analysis of the national VL program data from 2016 to 2020 at CPHL, to determine the association between low-level viraemia and viral non-suppression among PLHIV on ART in Uganda.

The study population comprised of PLHIV on ART with a suppressed VL (less than 1,000 copies/ml) done between January 2016 and December 2016 using plasma samples, and these were followed up to 31$^{st}$ December, 2020. Non-suppressed PLHIV on ART were excluded from this study. PLHIV with missing data for critical variables such as date of sample collection and results of the VL test for the follow-up years were also excluded from the study.

### The HIV viral load program description

The Ministry of Health established the national VL program in 2014, and this is housed and coordinated at CPHL. CPHL has a national VL reference testing laboratory which receives VL samples from all parts of the country through use of hub sample transport system [28]. CPHL has a comprehensive data centre that hosts the Laboratory Information Management System, which enables electronic delivery of VL results to health facilities through the electronic results download module [29].

The first VL test is done for PLHIV who have been on ART for 6 months, and then another VL test is done at 12 months if the results are suppressed. PLHIV with a VL result below 1,000 copies/ml have a suppressed VL and are routinely given adherence counselling in which they are encouraged to continue with their ART; and no other intervention is given [22]. PLHIV aged 18 years and above with a suppressed VL repeat a VL test once every year to monitor the efficacy of ART annually. PLHIV below 18 years who are suppressed do VL testing every 6 months. For PLHIV with a VL of 1,000 copies/ml or more, they are considered to be having a non-suppressed VL and they are offered monthly intensive adherence counselling (IAC) sessions for three months, after which a VL test is repeated in the fourth month to determine whether they have achieved VL suppression [22]. If the repeat VL result after IAC is still non-suppressed, this is considered as virologic failure provided non-adherence is ruled out, and a switch committee is convened and the patient is switched to another ART line [22].

## Data management and analysis

Routine viral load monitoring data for PLHIV on ART is archived into the laboratory information management system (LIMS) housed at CPHL. We developed and piloted a data abstraction checklist using a physical viral load result form to ensure that our checklist was comprehensive and captured all the relevant study data elements. We then extracted our study data from LIMS and the dataset included unique patient identifiers and other key variables such as age, sex, duration on ART, ART regimen, date of sample collection, type of VL sample collected, and the result of the VL test done. The dataset was extracted into a Microsoft spread sheet, cleaned by running filters to identify outliers and any other incomprehensible formats. We coded and removed duplicate records from the dataset before declaring it final for analysis. The remaining records were screened for inclusion into the study. Basing on the inclusion criteria, all PLHIV with non-suppressed results and all PLHIV with Dried Blood Spot (DBS) samples were excluded from the study.

The primary end point was viral non-suppression, and this was defined as a VL result of 1,000 copies/ml or more. The primary outcome was the proportions of PLHIV with VL non-suppression among the group with LLV and the group with a non-detectable VL. The secondary outcome was the time-to-non-suppression. The respective hazard ratios for non-suppression were also determined using Cox regression.

Our main exposure variable was level viraemia (LLV) defined as a VL result of at least 50 copies/ml but less than 1,000 copies/ml ($\geq$50 to <1,000 copies/ml). This was stratified into four categories basing on the level of viraemia, that is; 50 to 199, 200 to 399, 400 to 599, and 600 to 999 copies/ml. We further designated PLHIV with a VL less than 50 copies/ml (non-detectable VL) as the unexposed group.

## Statistical analysis

Using the approach of complete case analysis, descriptive statistics of participants with complete records were obtained where; continuous variables were summarised by mean (standard deviation) or median (Interquartile range) depending on the distribution, whereas categorical variables were summarised as frequencies and percentages. Multivariable logistic regression was used to determine the factors associated with low-level viraemia.

Cox proportional hazards regression model was used to determine the association between LLV and viral non-suppression, and schoenfeld residuals were used to test for the hazards assumption [30]. Hazard ratios, displayed as Kaplan-Meier estimators, were used to report the findings, comparing the relative risk of viral non-suppression for LLV with a non-detectable viral load. PLHIV who did not get viral non-suppression through out the follow-up period were right censored at the end of their last viral load test in 2020.

For the missing data, we determined the patterns and mechanisms of the missing data, and conducted multiple imputation using chained equations. We then ran model diagnostics to check whether the imputed results are similar to the observed results. We then derived the association between LLV and viral non-suppression for the inputed data. We conducted sensitivity analyses of both the observed and imputed results, and this showed relatively similar and consistent results. However we reported the results of complete case analysis in this study due to the large missingness of 72.2% of the data.

Stata version 14.0 was used for the statistical analysis of the data, and all the findings were reported with their respective 95% Confidence Interval levels.

## Ethical consideration

We acquired ethical approval from Makerere University School of Public Health Research Ethics Committee (SPHREC) to undertake this study, and the IRB approval number was SPH-

2021-144. We also acquired approval from the Uganda National Council for Science and Technology (UNCST), and the approval number was HS2008ES. Furthermore, permission to use the national viral load program data was sought from Ministry of Health. We received a waiver for the requirement of informed consent from SPHREC to use the archived national VL program data. All the program data was fully anonymized before we accessed it to ensure confidentiality of the patients.

## Results

### Trend of Low-level viraemia across the years

The national VL program data from 2016 to 2020 indicated increasing numbers of PLHIV with LLV from 2.0% in 2016 to 8.6% in 2020, and increasing suppression rates across the years, as shown in Fig 1 and Table 1.

### Association between low-level viraemia and non-suppression

Based on the inclusion criteria (Fig 2), a cohort of 17,783 PLHIV was followed from 2016 to 2020; of which 11,765 (66.2%) were female; 5,765 (32.4%) were male; and 253 (1.4%) had a missing sex result. The mean age for the study cohort was 33.8 years (SD—15.2), and the median follow-up time was 4.0 years (IQR 3.8–4.2). 15,741 (88.5%) PLHIV were on a first line ART regimen; 1,710 (9.6%) on a second line ART regimen; 53 (0.3%) on other regimen (third line regimens and any other salvage regimens), and 279 (1.6%) had a missing regimen; as shown in Table 2.

Low-level viraemia was identified in 1,466 PLHIV (8.2%); and of these, 836 (57.0%) were between 50 to 199 copies/ml; 288 (19.6%) were between 200 to 399 copies/ml; 129 (8.8%) were between 400 to 599 copies/ml; and 213 (14.6%) were between 600 to 999 copies/ml. Men had 1.3 times the risk of having LLV, as compared to women (risk ratio = 1.3, 95% CI: 1.2–1.4, p value < 0.001); and PLHIV on a second line ART regimen had 2.5 times the risk of having LLV, as compared to PLHIV with a first line ART regimen (risk ratio = 2.5, 95% CI: 2.2–2.8, p value < 0.001). Children had 3.1 times the risk of having LLV, as compared to adults (risk

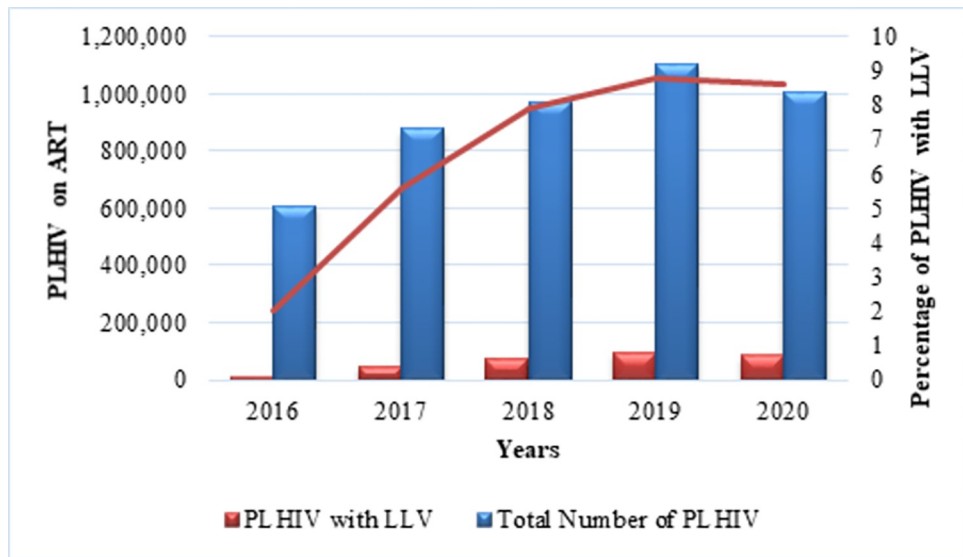

**Fig 1. Graph showing the trend of low-level viraemia from 2016 to 2020.**

**Table 1. Characteristics for the program data for the individual years from 2016 to 2020.**

| Characteristics | 2016 (n = 612,006) | 2017 (n = 917,460) | 2018 (n = 1,051,122) | 2019 (n = 1,230,992) | 2020 (n = 1,119,414) |
|---|---|---|---|---|---|
| **Mean Age (SD), years** | 36.3 (13.8) | 36.0 (14.0) | 36.2 (14.2) | 36.5 (14.4) | 37.0 (14.5) |
| **Gender** | | | | | |
| Female n (%) | 401, 974 (65.7) | 597,999 (65.2) | 686,989 (65.4) | 806,434 (65.5) | 737,825 (65.9) |
| Male n (%) | 202,380 (33.1) | 308,888 (33.7) | 351,937 (33.5) | 409,420 (33.3) | 368,974 (33.0) |
| Missing n (%) | 7,652 (1.2) | 10,573 (1.1) | 12,196 (1.1) | 15,138 (1.2) | 12,615 (1.1) |
| **Viral Suppression** | | | | | |
| Suppressed n (%) | 546,848 (89.4) | 787,950 (85.9) | 927,544 (88.2) | 1,097,658 (89.2) | 1,012,516 (90.5) |
| Non-Suppressed n (%) | 65,158 (10.6) | 129,510 (14.1) | 123,578 (11.8) | 133,334 (10.8) | 106,898 (9.5) |
| **ART regimen** | | | | | |
| First line regimen n (%) | 561,992 (91.8) | 847,117 (92.3) | 927,525 (88.2) | 1,091,537 (88.7) | 984,407 (87.9) |
| Second line regimen n (%) | 25,502 (4.2) | 41,236 (4.5) | 72,746 (6.9) | 87,566 (7.1) | 84,677 (7.6) |
| Other regimen n (%) | 1,016 (0.2) | 1,844 (0.2) | 16,749 (1.6) | 5,436 (0.4) | 8,600 (0.8) |
| Missing n (%) | 23,496 (3.8) | 27,263 (3.0) | 34,102 (3.3) | 46,453 (3.8) | 41,730 (3.7) |
| **Sample type** | | | | | |
| Plasma | 137,619 (22.5) | 318,440 (34.7) | 553,240 (52.6) | 617,106 (50.1) | 570,232 (50.9) |
| DBS | 474,387 (77.5) | 599,020 (65.3) | 497,882 (47.4) | 613,886 (49.9) | 549,182 (49.1) |

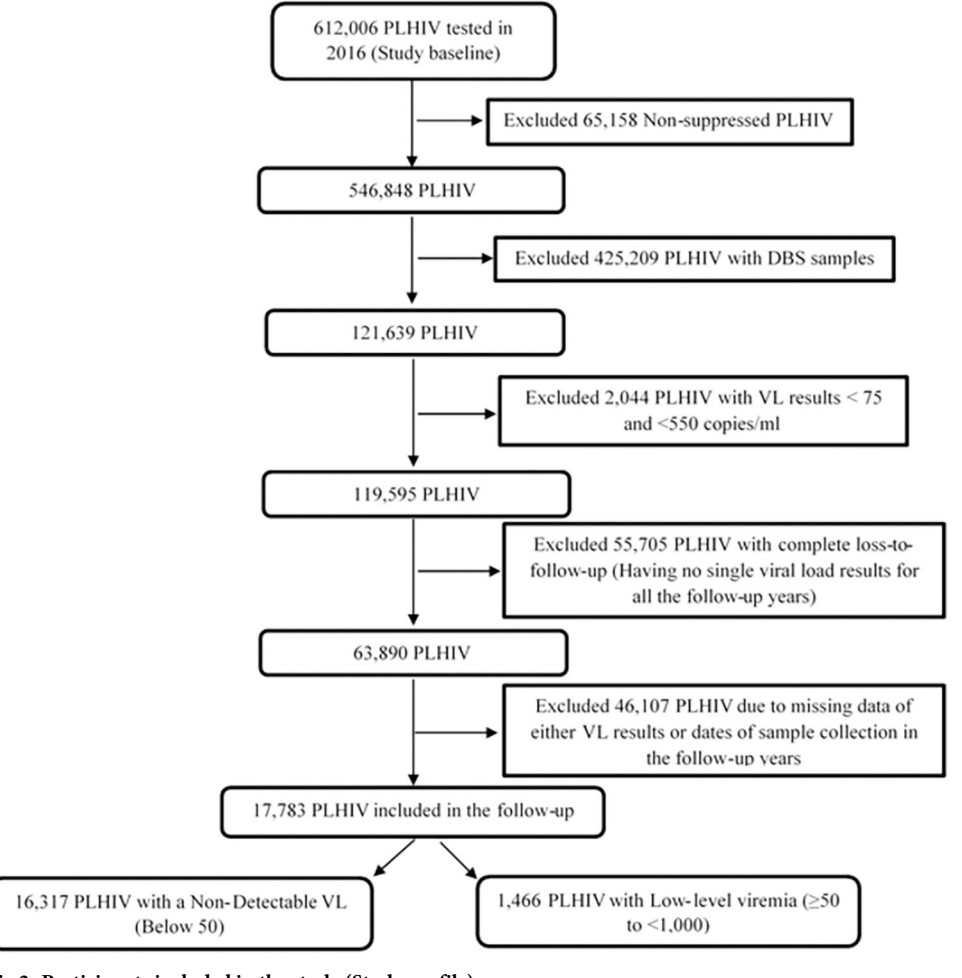

**Fig 2. Participants included in the study (Study profile).**

**Table 2. Characteristics of the study cohort at baseline in 2016.**

| Characteristics | (n = 17,783) |
|---|---|
| **Mean Age (SD), years** | 33.8 (15.2) |
| **Age disaggregation** | |
| Children n (%) | 3,399 (19.1) |
| Adults n (%) | 14,242 (80.1) |
| Missing n (%) | 142 (0.8) |
| **Gender** | |
| Female n (%) | 11,765 (66.2) |
| Male n (%) | 5,765 (32.4) |
| Missing n (%) | 253 (1.4) |
| **Level of viraemia** | |
| Non-Detectable (Below 50) n (%) | 16,317 (91.8) |
| Low-level viremia ($\geq$50 to <1,000) n (%) | 1,466 (8.2) |
| 50 to 199 copies/ml | 836 (57.0) |
| 200 to 399 copies/ml | 288 (19.6) |
| 400 to 599 copies/ml | 129 (8.8) |
| 600 to 999 copies/ml | 213 (14.6) |
| **ART regimen** | |
| First line regimen n (%) | 15,741 (88.5) |
| Second line regimen n (%) | 1,710 (9.6) |
| Other regimen n (%) | 53 (0.3) |
| Missing n (%) | 279 (1.6) |
| **Mean Duration on ART (SD), years** | 5.0 (3.3) |

ratio = 3.1, 95% CI: 2.9–3.4, p value < 0.001). Furthermore, young PLHIV were associated with LLV (mean age for PLHIV with LLV was 29.7 years, SD—16.4, as compared to the mean age for PLHIV with a non-detectable VL, which was 34.1 years, SD– 15.0).

A total of 1,730 PLHIV (9.7%) out of 17,783 PLHIV became non-suppressed during the follow-up period. 32.5% of the PLHIV with LLV (476 out of 1,466 PLHIV) became non-suppressed, as compared to 7.7% of the PLHIV (1,254 out of 16,317 PLHIV) with a non-detectable viral load who became non-suppressed during the follow-up period. PLHIV with LLV had 4.1 times the hazard rate of developing viral non-suppression, as compared to PLHIV with a non-detectable VL (adjusted hazard ratio was 4.1, 95%CI: 3.7 to 4.7, p < 0.000), after adjusting for age, sex, and ART regimen. High ranges of LLV were associated with increased hazard ratios, as shown in Table 3. The Kaplan-Meier estimators indicated that PLHIV with LLV had increased hazards of viral non-suppression, compared to PLHIV with a non-detectable viral load, and these hazards of viral non-suppression increased with increasing ranges of viraemia, as shown by Fig 3.

## Discussion

In this study, we assessed the association between low-level viraemia and viral non-suppression among PLHIV on ART in Uganda. To the best of our knowledge, this is the only such study to be conducted in Uganda, as of now. The findings of this study indicated that there are increasing proportions of PLHIV with LLV, from 2.0% in 2016 to 8.6% in 2020; and LLV was found to be associated with male sex, second line ART regimen and being of lower age. Furthermore, LLV was associated with a 4.1 times hazard rate of viral non-suppression as compared to a non-detectable VL.

**Table 3. The association between low-level viraemia and viral non-suppression as determined through cox proportional hazards analysis.**

| | Adjusted HR (95% CI) | p value |
|---|---|---|
| Non-Detectable (Below 50 copies/ml) | 1 (Reference) | |
| Low-level viremia (≥50 to <1,000) | 4.1 (3.7–4.6) | < 0.001 |
| 50 to 199 copies/ml | 2.4 (2.0–2.8) | < 0.001 |
| 200 to 399 copies/ml | 4.9 (4.0–5.9) | < 0.001 |
| 400 to 599 copies/ml | 10.1 (8.1–12.8) | < 0.001 |
| 600 to 999 copies/ml | 8.3 (6.9–10.1) | < 0.001 |
| Age: | | |
| **Adults** | 1 (Reference) | |
| **Children** | 2.8 (2.5–3.1) | < 0.001 |
| Sex: | | |
| Male | 1 (Reference) | |
| Female | 0.8 (0.7–0.9) | < 0.001 |
| ART Regimen: | | |
| First line regimen | 1 (Reference) | |
| Second line regimen | 1.6 (1.4–1.8) | < 0.001 |
| Other regimen | 0.8 (0.3–1.7) | 0.510 |

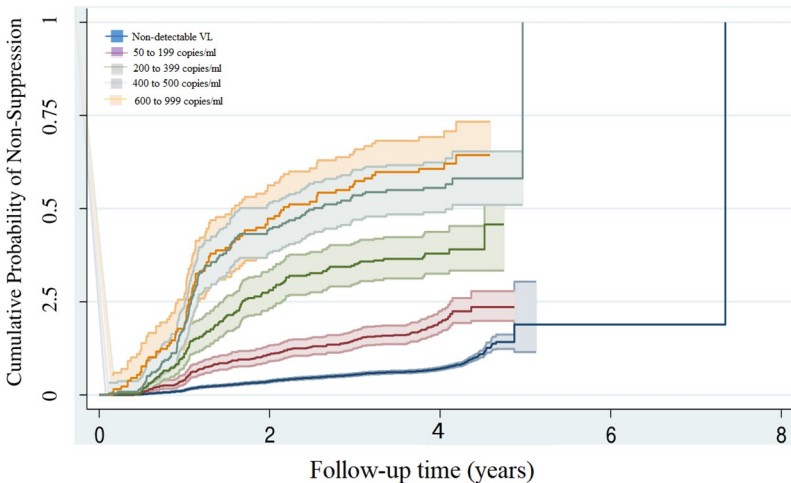

**Number at risk**

| | | | | | |
|---|---|---|---|---|---|
| Non-detectable VL | 16308 | 15710 | 7871 | 1 | 0 |
| 50 to 199 copies/ml | 836 | 745 | 308 | 0 | 0 |
| 200 to 399 copies/ml | 288 | 207 | 79 | 0 | 0 |
| 400 to 599 copies/ml | 129 | 68 | 28 | 0 | 0 |
| 600 to 999 copies/ml | 213 | 118 | 52 | 0 | 0 |

**Fig 3. Kaplan-Meier estimator showing the association between LLV and viral non-suppression.**

Our study builds on the existing literature about the emerging concern of LLV in Sub-Sahara Africa, and enriches the prevailing debate about the need to review the use of a threshold of 1,000 copies/ml to determine viral non-suppression among PLHIV on ART. The findings of our study are consistent with the findings of similar studies done in Sub-Saharan Africa, for instance a study in 4 African countries including Uganda showed high prevalences of viraemia and persistent LLV at 19.3% and 7.8% respectively among study participants [9]. This study also indicated that 57.5% of study participants with persistent viraemia had confirmed virologic failure. Similarly, the LLV study conducted in South Africa also indicated that LLV was in 23% of the study participants and LLV was associated with increased hazards of virological failure (hazard ratio of 2.6, 95% CI: 2.5–2.8; p<0.0001) [10].

Our findings indicate four main aspects that are important to note. The first aspect is that there are increasing PLHIV with LLV in Uganda over the years, from 2.0% in 2016 to 8.6% in 2020. This implies that there is an urgent need in the country to assess for the cause of this rising trend of LLV, and institute strategies to halt it. Secondly, LLV has been shown to be associated with male sex, among other factors; and this is not a surprise because men have been associated with poor health seeking behaviour, as compared to women [31]. Furthermore, different barriers like self-denial, HIV related stigma, conflicting priorities of having to work to get money to look after the family, alcohol ingestion, and perceived medication effects are key in hindering men from seeking HIV-related care, which may contribute to poor treatment outcomes like LLV [31–33]. This therefore calls for concerted efforts to address these barriers, which can consequently improve treatment outcomes in males, including reduction of LLV. Furthermore, our findings show that children have increased rates of LLV, as compared to adults. This is in line with previous studies which have shown increased rates of other poor treatment outcomes like virologic failure and drug resistance among children [16, 34, 35]. These poor treatment outcomes have been associated with non-disclosure, HIV related stigma, and ART-induced side effects, among others [16, 36, 37]. Hence the need to devote efforts to address these challenges to reduce the rates of LLV and other poor treatment outcomes among children. Lastly, LLV has been associated with a 4.1 times the rate of viral non-suppression as compared to a non-detectable VL and it is important to note that this rate increases with high ranges of LLV. This indicates that higher ranges of viraemia are highly associated with viral non suppression and virologic failure, as also shown in several other studies [10], hence Uganda should consider a policy to revise the VL non-suppression threshold to at least 400 copies/ml, putting the public health perspective in mind. Though the ideal would be 200 copies/ml as recommended by CDC and IAPAC [19, 20].

## Strengths and limitations

Among the various strengths of our study is that we followed up a large number of PLHIV for a long period of time of upto 5 years, in determining the association between LLV and VL non-suppression.

Our study had several limitations, the first being that this was an observational retrospective cohort analysis using the national program data for viral load testing. As a result, there was significant missing data for key variables like the date of sample collection and viral load results for the follow-up years (2017, 2018, 2019 and 2020). However we conducted a comprehensive missing data analysis. The percentage missingness was 72.2% whereby 46,107 out 63,890 PLHIV missed one of the key variables (either a date of sample or viral load result) in the follow-up years. The missing data followed a non-monotonic and multivariate pattern. Despite this pattern, it is important to note that the PLHIV who missed a date of sample collection also missed a viral load result. To determine the mechanism of the missingness, we ran the Little's

MCAR Test and the p value was < 0.001, hence the data was not missing completely at random (MCAR). We then did a logistic regression to test whether the data was missing at random (MAR), and the p value was < 0.001, hence the data was MAR. We then conducted multiple imputation using chained equations, after which we ran the model diagnostics to check whether the imputed results are similar to the observed results. Results from multiple imputation indicated that PLHIV with LLV had 6.1 times the hazard rate of developing viral non-suppression, as compared to PLHIV with a non-detectable VL (adjusted hazard ratio was 6.1, 95% CI: 5.4 to 6.8, p < 0.001). This was comparable and had similar conclusions with the findings of complete case analysis. Important to note that the standard multiple imputation approach did not account for the intra-cluster correlations of the different ranges of viraemia. Further sensitivity analysis of both the observed and imputed results gave consistent results, and due to the large missingness of 72.2%, we decided to report the results of the complete case analysis in this study.

Secondly, the other limitation is that we used the national VL program data, and this lacked results for HIV drug resistance testing, which would have provided a further understanding of LLV in this study. The national program data also lacked data on other key variables like co-morbidities which could be potential confounders. Furthermore, different assays and techniques could have been used in conducting the VL testing for the different PLHIV over the years, which could create a threat of measurement bias. However in this study, we only included participants whose VL tests were done on plasma samples, since techniques that use plasma samples give more distinct results, as compared to those that use DBS samples.

## Conclusion

Our study indicated that PLHIV with LLV had 4.1 times the hazard rate of developing viral non-suppression, as compared to PLHIV with a non-detectable VL, and that this rate increased with elevated ranges of LLV. These findings indicate that there is an urgent need to review the VL testing alogarithm in Uganda, and also institute interventions such as intensive adherence counselling to manage LLV among PLHIV on ART, as per the recent 2021 WHO recommendations.

## Supporting information

**S1 Dataset. Dataset for the study cohort from 2016 to 2020.**
(XLSX)

## Acknowledgments

We extend our appreciation to Mr. Obuya Emmanuel, Mrs. Nabukenya Miriam Bamwita, Mr. Kiwanuka Julius, Mr. Ssemugabo Charles, Ms. Mary Nakafeero and Mrs. Esther Bayiga for the devoted efforts invested in the study's data management and statistical analysis processes. Thank you very much.

## Author Contributions

**Conceptualization:** Nicholus Nanyeenya, Larry William Chang, Noah Kiwanuka, Damalie Nakanjako, Gertrude Nakigozi, Fredrick Makumbi.

**Data curation:** Esther Nasuuna, Simon P. S. Kibira, Susan Nabadda, Charles Kiyaga.

**Formal analysis:** Nicholus Nanyeenya, Noah Kiwanuka, Gertrude Nakigozi, Simon P. S. Kibira, Susan Nabadda, Charles Kiyaga, Fredrick Makumbi.

**Funding acquisition:** Nicholus Nanyeenya, Larry William Chang, Fredrick Makumbi.

**Methodology:** Nicholus Nanyeenya.

**Project administration:** Susan Nabadda, Charles Kiyaga.

**Software:** Esther Nasuuna.

**Supervision:** Noah Kiwanuka, Damalie Nakanjako, Susan Nabadda, Fredrick Makumbi.

**Writing – original draft:** Nicholus Nanyeenya, Esther Nasuuna, Fredrick Makumbi.

**Writing – review & editing:** Nicholus Nanyeenya, Larry William Chang, Noah Kiwanuka, Esther Nasuuna, Damalie Nakanjako, Gertrude Nakigozi, Simon P. S. Kibira, Susan Nabadda, Charles Kiyaga, Fredrick Makumbi.

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
