## [Decision Letter · Decision Letter 0]

27 Sep 2022

PONE-D-22-22579The association between low-level viraemia and subsequent viral non-suppression among people living with HIV/AIDS on antiretroviral therapy in UgandaPLOS ONE

Dear Dr. Nanyeenya,

Thank you for submitting your manuscript to PLOS ONE. After careful consideration, we feel that it has merit but does not fully meet PLOS ONE’s publication criteria as it currently stands. Therefore, we invite you to submit a revised version of the manuscript that addresses the points raised during the review process. Suggestions to improve the manuscripts can be taken from the comments of the two reviewers. There are critical issues but they can be addressed. Kindly consider their comments and suggestions and revise as appropriate

We look forward to receiving your revised manuscript.

Kind regards,

Chika Kingsley Onwuamah, Ph.D.

Academic Editor

PLOS ONE

Journal Requirements:

2. In the ethics statement in the manuscript and in the online submission form, please provide additional information about the patient records/samples used in your retrospective study. Specifically, please ensure that you have discussed whether all data/samples were fully anonymized before you accessed them and/or whether the IRB or ethics committee waived the requirement for informed consent. Please clearly state which ethics committee provided the waiver for the requirement of informed consent.

Reviewers' comments:

Reviewer's Responses to Questions

**Comments to the Author**

1. Is the manuscript technically sound, and do the data support the conclusions?

Reviewer #1: Partly

Reviewer #2: Yes

2. Has the statistical analysis been performed appropriately and rigorously? 

Reviewer #1: No

Reviewer #2: Yes

3. Have the authors made all data underlying the findings in their manuscript fully available?

Reviewer #1: Yes

Reviewer #2: Yes

4. Is the manuscript presented in an intelligible fashion and written in standard English?

Reviewer #1: Yes

Reviewer #2: Yes

5. Review Comments to the Author

Reviewer #1: Abstract;

Background: The introductory statement is not in tandem with the aim and objective of the study. One is rather worried about HIV drug resistance in patients with low level viraemia.

Introduction;

Line - 96-97 - The introduction is satisfactory, however there are repetitions which was noticed on line 66 -68

Line 107-108 – The statement should read that we aimed to determine but not to understand the association between low level viraemia……

Line - 151-153 - Exposed to what? Is this the right word to use?

Methodology. Well presented and sufficient to allow reproducibility

Results

Line - 194 Fig 1 Not displayed

Line - 224 Fig 2 Not displayed

Line - 240 Table 3 – Include %

The manuscript did not mention the ARV regimen for the 1st/2nd line ARVs

The authors did not mention other cofounders eg comorbities in the patients/data reviewed that can affect viral suppression

The socio demographic characteristics of the reviewed data was not highlighted.

The data did not provide the age disaggregation except by gender

What do you mean by “other regimen” in table 3?

Statistical analysis is okay but not rigorously performed.

Discussion

Line - 281-282 – Repetition of statement

Line - 285 -286 - What about the fact that men may be engaged with some activities to provide for the family and may forget their medications. What about denial state. Men may have the preponderance to indulge in alcohol ingestion and may forget or interference with medications

Line - 289 -291 - Repeating the result in the discussion is not acceptable.

Line - 302 -312 - Why so much emphasis and statistical analysis on missing data which does not form part of the data presented

Line - 318-319 – Is this statement a limitation or a strength?

Line - 321-323. However, in this study, we only included participants whose VL tests were done on plasma samples, since techniques that use plasma samples give more distinct results, as compared to those that use DBS samples

Another limitation is that children were not included in the study and from reports the children are more predisposed to LLV and non-viral suppression

Conclusion

327-329 In response to the last sentence in your conclusion, would there be a need to review the drug regimen? Why WHO recommendations, Does Uganda not have National guidelines on ART? The recommendations need to be apt.

References

Authors should review reference 19 and other references and ensure it align with the journal instructions.

The fonts and characters are different what is in the text

General Comments

The authors have generally made significant effort to describe their findings on patients on ART with LLV in Uganda. The sample size is large enough to draw an inference. They have also highlighted their strength and limitations. However, I observed significant limitations of this study which obviously excluded children who are less than 18 years. Is it that within this period of study children and early adolescents did not present with low level viraemia.

There were repetitions in the write up and the figures were not in place though efforts should have been made to refer the reviewers to the appendix.

Generally, the discussion did not critically review the findings and critic those findings in relation to other studies and available literatures. The authors were instead discussing missing data. This and other queries must be addressed.

Reviewer #2: TITLE: THE ASSOCIATION BETWEEN LOW-LEVEL VIRAEMIA AND SUBSEQUENT VIRAL NON-SUPRESSION AMONG PEOPLE LIVING WITH HIV/AIDS ON ANTIRETROVIRAL THERAPHY IN UGANDA

SUMMARY OF THE PAPER

This article describes a retrospective cohort study on the association between low-level HIV viraemia and subsequent virologic non-suppression. It is a retrospective cohort study of 17,783 people living with HIV out of which 1,466 had low-level viraemia and the remaining had undetectable viral load. It was a five year follow up study (2016 - 2020). The study defined undetected viral load as viral load below 50 while low-level viral load was taken as between 51 copies and 1000 copies. The study found that during the follow up period, the incident proportion of persons with low-level viraemia increased from 2% to 8.6% and that persons with low-level viral load had 4.1 times the risk of failure to achieve viral suppression compared to those with undetected viral load. Low-level viraemia was also associated with the male gender and being on second line antiretroviral theraphy. The authors of the paper established on literature review that low-level viraemia was associated with antiretroviral drug resistance and virologic failure. The researchers employed the approach of complete case analysis and multivariate logistic regression to establish an association between low-level viraemia and virologic non-suppression. They also used the cox proportional hazard regression model to identify the association between low-level viraemia and viral suppression. Study limitations included the fact that a large part of the cohort about 72% had missing data that was essential to the study and that resistance testing was not available for the cohort.

ASSESSMENTS

This study is important for giving the reader an insight into the challenges of managing people living with HIV in Uganda. Although the title describes what was actually carried out in the study , it does not capture an important aspect of what the researchers tried to contribute. The researchers were obviously concerned and objected to a continued policy of accepting a viral load of 1000 as threshold for satisfactory treatment response. And related to this, the need for greater attention to be paid to persons with low-level viraemia especially the young male on second line antiretroviral theraphy.

This aspect of the manuscript is one of the major strengths i.e pointing out the weakness of current policy and providing data to challenge it. The authors however did not make a strong enough connection between their findings and the need for a review of this policy. The major weakness of this study is the treatment of missing data that was essential to the study.

IMPRESSION OF PAPER/RECOMMENDATION

The overall arrangement of the paper is good, the language is sytematic and scientific, the evidence was well brought out and the conclusions matched the evidence. However, the recommendation which had to do with the need for policy change was not well argued. The authors make an overall good impression and the paper is worthy of dissemination in peer reviewed journals.It should be Accepted with revisions as follows.

MAJOR ISSUES

1.[Line 304 to 316] Author needs to clarify the treatment of missing data to show the proportion of people with missing date of sample and the proportion of people with missing viral load result because the effect of a missing viral load result would not necessarily be the same as the effect of a missing sample date.

MINOR ISSUES

1.Table 3 does not include age which is already identified as a significant factor.

2. The paper did not give the reader the rational for using ‘virologic non-suppression’ rather than ‘virologic failure’.

REVIEWER: Dr. Harry OHWODO

6. PLOS authors have the option to publish the peer review history of their article (what does this mean?). If published, this will include your full peer review and any attached files.

Reviewer #1: No

Reviewer #2: **Yes: **Harry Ohwodo

---

## [Author Response · Author response to Decision Letter 0]

14 Oct 2022

First and foremost, we greatly and humbly thank you all for taking off time to carefully read our manuscript and giving us very insightful review comments. We greatly believe that addressing these comments has been very key in improving our manuscript. Thank you very much once again.

We hereby humbly submit the responses to the different review comments raised, as shown below;

1. ACADEMIC EDITOR

Thank you for submitting your manuscript to PLOS ONE. After careful consideration, we feel that it has merit but does not fully meet PLOS ONE’s publication criteria as it currently stands. Therefore, we invite you to submit a revised version of the manuscript that addresses the points raised during the review process.

Thank you very much for this comment. We have read through PLOS ONE’s style requirements and revised the manuscript to meet these requirements. Thank you very much.

b) In the ethics statement in the manuscript and in the online submission form, please provide additional information about the patient records/samples used in your retrospective study. Specifically, please ensure that you have discussed whether all data/samples were fully anonymized before you accessed them and/or whether the IRB or ethics committee waived the requirement for informed consent. Please clearly state which ethics committee provided the waiver for the requirement of informed consent.

This is a key comment. Thank you very much. We have revised the ethics statement in the manuscript as shown in lines 179 to 185, and in the online submission to incorporate these key aspects highlighted in this comment. 

c) We note that the grant information you provided in the ‘Funding Information’ and ‘Financial Disclosure’ sections do not match. 

Thank you for this observation. This has been addressed. Thank you very much.

Thank you very much for this. We have addressed this, as per your guidance.

2. REVIEWER #1

a) Background: The introductory statement is not in tandem with the aim and objective of the study. One is rather worried about HIV drug resistance in patients with low level viraemia.

Thank you very much for this comment. We have revised the introductory statement, as shown in lines 94 to 96.

b) Introduction;

i. Line - 96-97 - The introduction is satisfactory, however there are repetitions which was noticed on line 66 -68

Thank you very much. This has been addressed and revised as shown in lines 63 to 65. 

ii. Line 107-108 – The statement should read that we aimed to determine but not to understand the association between low level viraemia……

Thanks for this guidance. This has been revised as shown in lines 105. 

iii. Line - 151-153 - Exposed to what? Is this the right word to use?

Thank you very much. The exposure was low-level viraemia. However, following this comment, the sentence has been revised to make it clear, as shown in lines 150 to 151.

c) Methodology. Well presented and sufficient to allow reproducibility

Thank you very much.

d) Results

i. Line - 194 Fig 1 Not displayed

Thank you very much. We desired to place all the figures in the manuscript however following the guidance from the journal, figure 1 was separately uploaded into the system. Thanks very much.

ii. Line - 224 Fig 2 Not displayed

Thank you very much. We also desired to place this figure in the manuscript however following the guidance from the journal, this figure was also separately uploaded into the system. Thanks very much.

iii. Line - 240 Table 3 – Include %

Thank you for this comment. We humbly request you to note that Table 3 represents the adjusted hazards ratios, which do not require %. However we have noted that we had included ‘n (%)’ on the ‘other regimen’ by mistake, and we have corrected this, as shown in Table 3. Thanks indeed for this comment.

iv. The manuscript did not mention the ARV regimen for the 1st/2nd line ARVs

Thank you very much. It would really be great to mention the ARV regimens, however Uganda has over 20 ARV regimens for the first and second line, which made mentioning them not so feasible. Therefore, we decided to follow literature where many other similar articles mainly stratify ARVs up to the level of first and second line only, without mentioning the respective regimens. Thank you very much.

v. The authors did not mention other cofounders e.g. comorbities in the patients/data reviewed that can affect viral suppression

This is a very insightful comment. Thank you very much. We used national VL program data which did not have complete data for some variables like comorbidities. However in the data analysis, we addressed confounding by stratification, as evidenced by the reported adjusted hazards ratios in Table 3. Following this comment, a discussion of using national program data with missing data on probable confounders has been added to the Discussion section, as shown by line 314 to 315. Thank you very much. 

vi. The socio demographic characteristics of the reviewed data was not highlighted.

Thank you for this comment. We humbly request you to note that Table 1 on line 205 shows the socio demographic characteristics of the reviewed data for the VL program for the individual years from 2016 to 2020. Table 2 on line 232 also shows the socio demographic characteristics of the Study Cohort at baseline in 2016. Thank you very much.

vii. The data did not provide the age disaggregation except by gender

Thank you very much for this perfect comment. We have realized that including the age disaggregation would add much value to the manuscript. Following the guidance from this comment, we have repeated the data analysis, to include the disaggregation by age (children versus adults). We have revised the manuscript to report on the children, as shown in lines 220 to 223, Table 2 on line 232, and Table 3 on line 244. We have also included this in our discussion as shown in lines 277 to 282. Greatly thank you for this key comment.

viii. What do you mean by “other regimen” in table 3?

Thanks very much. The national ART guidelines in Uganda refer to any third line regimen or any other salvage ART combination (regimen) used in management of drug resistance, as ‘other regimen’. This has been further clarified in the manuscript, as shown in lines 213 to 214.

e) Statistical analysis is okay but not rigorously performed.

Thanks very much.

f) Discussion

i. Line - 281-282 – Repetition of statement

Thanks very much. This comment is well noted. We are humbly informing you that these are two different studies, however this comment is very true because it looked like a repetition. We have revised these statements in the manuscript following this comment to ensure that they do not look like a repetition, as shown in line 263 to 265.

ii. Line - 285 -286 - What about the fact that men may be engaged with some activities to provide for the family and may forget their medications. What about denial state. Men may have the preponderance to indulge in alcohol ingestion and may forget or interference with medications

Thanks for this comment. We humbly note that this is very useful guidance, and we have done further literature review to incorporate these factors. The manuscript has been revised following this guidance, as shown in lines 272 to 276.

iii. Line - 289 -291 - Repeating the result in the discussion is not acceptable.

Thank you very much. This has been revised in the manuscript as shown in lines 282 to 284. Thanks.

iv. Line - 302 -312 - Why so much emphasis and statistical analysis on missing data which does not form part of the data presented

Thanks very much. We humbly note this comment however our team feels that it is important to include these details in the discussion, to show how we managed missing data, since it was a key weakness of our study. However as per your guidance in the other comments, we have improved the other sections of the discussion to make it better. Thank you.

v. Line - 318-319 – Is this statement a limitation or a strength?

Thanks. This statement is a limitation. However this statement has been revised in the manuscript to make it clear, as shown in line 312. Thank you very much.

vi. Line - 321-323. However, in this study, we only included participants whose VL tests were done on plasma samples, since techniques that use plasma samples give more distinct results, as compared to those that use DBS samples

Thank you very much. We used this statement to explain how we tried to solve the limitation of different assays used for viral load testing, which could lead to measurement bias. Thank you very much.

g) Another limitation is that children were not included in the study and from reports the children are more predisposed to LLV and non-viral suppression

Thanks very much for this insightful comment. The inclusion criteria for the study was people living with HIV (PLHIV) on ART with a suppressed viral load, done using a plasma sample. By this criteria, children were included into the study. However we acknowledge that we had not extensively included them in the data analysis and had not reported on them. We have done the data analysis again, and reported on children, as shown in lines shown in lines 220 to 223, Table 2 on line 232, and Table 3 on line 244. We have also included this in our discussion as shown in lines 277 to 282. Thank you.

h) Conclusion:

i. 327-329 In response to the last sentence in your conclusion, would there be a need to review the drug regimen? Why WHO recommendations, Does Uganda not have National guidelines on ART? The recommendations need to be apt.

Thank you very much for this. For this study, we think that there may be no need to review the drug regimens since Dolutegravir-based regimens have just been rolled out within the previous 3 years in Uganda, and other sub-Sahara African countries. However we agree that further research and studies should be done to review drug regimens.

Thanks for hinting about the WHO recommendation. In 2021, WHO recommended use of intensive adherence counselling (IAC) in management of low-level viraemia (LLV). However many countries including Uganda are still very reluctant to initiate IAC for LLV, and it is actually not yet incorporated in the Ugandan HIV consolidated guidelines. And that is why we emphasize that interventions like IAC should be instituted in management of LLV, basing on the WHO recommendation. However following this comment, we have clarified the last sentence to indicate the year of the WHO recommendation, as shown in line 332.

i) References

i. Authors should review reference 19 and other references and ensure it align with the journal instructions.

Thanks for this comment. This has been revised in the manuscript, as shown in lines 382 to 383. Thank you very much.

ii. The fonts and characters are different what is in the text

Thank you very much. This has been addressed in manuscript, as shown in the reference section.

j) General Comments

The authors have generally made significant effort to describe their findings on patients on ART with LLV in Uganda. The sample size is large enough to draw an inference. They have also highlighted their strength and limitations. However, I observed significant limitations of this study which obviously excluded children who are less than 18 years. Is it that within this period of study children and early adolescents did not present with low level viraemia.

There were repetitions in the write up and the figures were not in place though efforts should have been made to refer the reviewers to the appendix.

Generally, the discussion did not critically review the findings and critic those findings in relation to other studies and available literatures. The authors were instead discussing missing data. This and other queries must be addressed.

Thank you very much for all the comments raised. The manuscript has been revised following the guidance from the comments. The data has been re-analysed to include the children, and the discussion has also been improved, in addition to addressing the other comments. Thank you very much.

3. REVIEWER #2

a) Summary of the paper

This article describes a retrospective cohort study on the association between low-level HIV viraemia and subsequent virologic non-suppression. It is a retrospective cohort study of 17,783 people living with HIV out of which 1,466 had low-level viraemia and the remaining had undetectable viral load. It was a five year follow up study (2016 - 2020). The study defined undetected viral load as viral load below 50 while low-level viral load was taken as between 51 copies and 1000 copies. The study found that during the follow up period, the incident proportion of persons with low-level viraemia increased from 2% to 8.6% and that persons with low-level viral load had 4.1 times the risk of failure to achieve viral suppression compared to those with undetected viral load. Low-level viraemia was also associated with the male gender and being on second line antiretroviral therapy. The authors of the paper established on literature review that low-level viraemia was associated with antiretroviral drug resistance and virologic failure. The researchers employed the approach of complete case analysis and multivariate logistic regression to establish an association between low-level viraemia and virologic non-suppression. They also used the cox proportional hazard regression model to identify the association between low-level viraemia and viral suppression. Study limitations included the fact that a large part of the cohort about 72% had missing data that was essential to the study and that resistance testing was not available for the cohort.

b) Assessments

This study is important for giving the reader an insight into the challenges of managing people living with HIV in Uganda. Although the title describes what was actually carried out in the study, it does not capture an important aspect of what the researchers tried to contribute. The researchers were obviously concerned and objected to a continued policy of accepting a viral load of 1000 as threshold for satisfactory treatment response. And related to this, the need for greater attention to be paid to persons with low-level viraemia especially the young male on second line antiretroviral therapy.

This aspect of the manuscript is one of the major strengths i.e pointing out the weakness of current policy and providing data to challenge it. The authors however did not make a strong enough connection between their findings and the need for a review of this policy. The major weakness of this study is the treatment of missing data that was essential to the study.

c) Impression of paper/recommendation

The overall arrangement of the paper is good, the language is systematic and scientific, the evidence was well brought out and the conclusions matched the evidence. However, the recommendation which had to do with the need for policy change was not well argued. The authors make an overall good impression and the paper is worthy of dissemination in peer reviewed journals. It should be Accepted with revisions as follows.

d) Major issues

i. [Line 304 to 316] Author needs to clarify the treatment of missing data to show the proportion of people with missing date of sample and the proportion of people with missing viral load result because the effect of a missing viral load result would not necessarily be the same as the effect of a missing sample date.

Thanks very much for raising this very critical comment. Largely, PLHIV with a missing date, also missed a viral load result for example in 2017, 61.5% of PLHIV missed the date of sample collection while 58.7% missed a viral load result. In 2018, 63.2% of the PLHIV missed both the date of sample collection and the viral load result. In 2019, 66.2% of the PLHIV missed both, while in 2020, 70.3% of the PLHIV missed both the date of sample collection and viral load results. This indicates a loss-to-follow up of these participants. We imputed all these missing variables to assess the impact of the missingness, and this is why we described the full steps that we used to manage the missing data in the discussion. We have made further clarification in the manuscript about this, as shown in line 298 to 299. Once again, thank you for highlighting this comment.

d) Minor issues

i. Table 3 does not include age which is already identified as a significant factor.

Thank you for this key comment. We acknowledge that we had missed out data analysis on age disaggregation (children versus adults). However this has been addressed as shown in Table 3 on line 244.

ii. The paper did not give the reader the rational for using ‘virologic non-suppression’ rather than ‘virologic failure’.

Thank you very much for this comment. We had desired to use virologic failure in this study, however we used the national viral load program data, which had a limitation of a lot of missing data, and hence we would not ascertain the virologic failure very well. However virologic non-suppression is a major predictor of virologic failure, according to literature, and actually key stakeholders like CDC define virologic failure as the inability to achieve or maintain suppression of viral replication to HIV-RNA level <200 copies/mL, which is the same as virologic non-suppression (https://clinicalinfo.hiv.gov/en/guidelines/hiv-clinical-guidelines-adult-and-adolescent-arv/virologic-failure).

Once again, thank you very much for the comments.

---

## [Editor Report · Decision Letter 1]

8 Dec 2022

The association between low-level viraemia and subsequent viral non-suppression among people living with HIV/AIDS on antiretroviral therapy in Uganda

PONE-D-22-22579R1

Dear Dr. Nanyeenya,

We’re pleased to inform you that your manuscript has been judged scientifically suitable for publication and will be formally accepted for publication once it meets all outstanding technical requirements.

Kind regards,

Chika Kingsley Onwuamah, Ph.D.

Academic Editor

PLOS ONE
---

## [Editor Report · Acceptance letter]

4 Jan 2023

PONE-D-22-22579R1 

The association between low-level viraemia and subsequent viral non-suppression among people living with HIV/AIDS on antiretroviral therapy in Uganda 

Dear Dr. Nanyeenya:

I'm pleased to inform you that your manuscript has been deemed suitable for publication in PLOS ONE. Congratulations! Your manuscript is now with our production department. 

Kind regards, 

on behalf of

Dr. Chika Kingsley Onwuamah 

Academic Editor

PLOS ONE